# Molecular Insights into the Oxygen-Sensing Pathway and Erythropoietin Expression Regulation in Erythropoiesis

**DOI:** 10.3390/ijms22137074

**Published:** 2021-06-30

**Authors:** Jana Tomc, Nataša Debeljak

**Affiliations:** 1Medical Centre for Molecular Biology, Institute of Biochemistry and Molecular Genetics, Faculty of Medicine, University of Ljubljana, 1000 Ljubljana, Slovenia; jana.tomc@mf.uni-lj.si; 2Centre for Functional Genomics and Bio-Chips, Institute of Biochemistry and Molecular Genetics, Faculty of Medicine, University of Ljubljana, 1000 Ljubljana, Slovenia

**Keywords:** erythropoiesis, oxygen-sensing pathway, hypoxia-inducible factor, erythropoietin, molecular mechanisms, post-translational modifications, protein interactions, nuclear transport, transcriptional regulation

## Abstract

Erythropoiesis is regulated by several factors, including the oxygen-sensing pathway as the main regulator of erythropoietin (EPO) synthesis in the kidney. The release of EPO from the kidney and its binding to the EPO receptor (EPOR) on erythrocyte progenitor cells in the bone marrow results in increased erythropoiesis. Any imbalance in these homeostatic mechanisms can lead to dysregulated erythropoiesis and hematological disorders. For example, mutations in genes encoding key players of oxygen-sensing pathway and regulation of EPO production (HIF-EPO pathway), namely *VHL*, *EGLN*, *EPAS1* and *EPO*, are well known causative factors that contribute to the development of erythrocytosis. We aimed to investigate additional molecular mechanisms involved in the HIF-EPO pathway that correlate with erythropoiesis. To this end, we conducted an extensive literature search and used several in silico tools. We identified genes encoding transcription factors and proteins that control transcriptional activation or repression; genes encoding kinases, deacetylases, methyltransferases, conjugating enzymes, protein ligases, and proteases involved in post-translational modifications; and genes encoding nuclear transport receptors that regulate nuclear transport. All these genes may modulate the stability or activity of HIF2α and its partners in the HIF-EPO pathway, thus affecting EPO synthesis. The theoretical information we provide in this work can be a valuable tool for a better understanding of one of the most important regulatory pathways in the process of erythropoiesis. This knowledge is necessary to discover the causative factors that may contribute to the development of hematological diseases and improve current diagnostic and treatment solutions in this regard.

## 1. Introduction

Erythropoiesis is a process of red blood cell (RBC) production which is regulated by a negative feedback loop of tissue oxygenation and hormone erythropoietin (EPO) synthesis. Expression of EPO in the kidney in low tissue oxygenation is induced by the hypoxia-inducible transcription factor (HIF) via the oxygen-sensing pathway. EPO is thereafter released to the bloodstream and binds to the EPO receptor (EPOR) on red cell progenitors in the bone marrow, triggering erythropoiesis and resulting in compensation of tissue hypoxia [1]. Based upon the key molecular players involved in this process the pathway is named also the HIF-EPO pathway. The production of RBC involves a commitment of pluripotent hematopoietic stem cells to the erythroid lineage and their progression to terminally differentiated erythrocytes. The earliest progenitors committed to the erythroid lineage are burst-forming unit-erythroid cells and their proliferation is promoted by several factors, including stem cell factor (SCF) and insulin-like growth factor I (IGF1). Colony-forming unit-erythroid cells (CFU-E) are the next stage of erythroid progenitors. CFU-E divide rapidly and are highly responsive to EPO. These cells give rise to proerythroblasts which become basophilic, polychromatophilic, and orthochromatophilic erythroblasts. CFU-E, proerythroblasts, and early basophilic erythroblasts express EPOR on their surface, and binding of EPO to EPOR activates the downstream signals, preventing the cells from apoptosis and stimulating their proliferation, differentiation, and maturation. Orthochromatophilic erythroblasts undergo enucleation and form reticulocytes which are released into the bloodstream, and there they develop into mature erythrocytes [2]. The oxygen capacity of the blood is thus enhanced and delivery of oxygen to tissues increased, thereby completing the negative feedback loop.

Any imbalance in these homeostatic mechanisms can result in hematological disorders, such as erythrocytosis. Therefore, the understanding of the main regulatory pathways in the process of erythropoiesis is necessary to discover the causative factors, which can contribute to the development of hematological disorders, and improve the current diagnostic solutions in this regard. 

The aim of this study was to investigate the molecular mechanisms involved in the oxygen-sensing pathway and regulation of erythropoietin production that correlate with erythropoiesis. We reviewed factors acting on the level of (i) transcriptional regulation and (ii) protein interactions, including post-translational modifications and nuclear transport. All these factors alone or cumulatively modulate the stability or activity of the main players in the HIF-EPO pathway, and thus influence the EPO synthesis and final RBC production. In this respect, we conducted an extensive literature search and used several in silico tools, such as Reactome [3], String [4], UniProt [5], GeneCards [6], Human Protein Atlas [7], NCBI [8], and HGNC [9] databases. The literature research was performed using keywords such as oxygen-sensing pathway, hypoxia inducible factor, HIF, HIF1A, HIF2A, EPAS1, PHD2, EGLN1, VHL, erythropoietin, EPO, EPO synthesis, hypoxia, erythropoiesis, red blood cell production, and erythrocytosis. Gene names were unified according to the NCBI Gene database.

## 2. Main Players in the Oxygen-Sensing Pathway and EPO Expression Regulation

EPO is a key regulator of erythropoiesis and its constant production is necessary to maintain the daily renewal of RBC [10]. The major site of EPO production is specialized interstitial fibroblasts in the kidney, while a small amount can also be produced in hepatocytes [11]. During stress erythropoiesis, EPO expression can also be induced by bone marrow osteoblasts [12]. EPO expression is regulated by hypoxia-inducible transcription factors (HIFs) which consist of O_2_-dependent α subunit (HIFα) and constitutively expressed β subunit (HIF1β or ARNT) [1]. In normal oxygen conditions, the HIFα subunit is continuously transcribed, translated, and readily degraded by the ubiquitin-proteasome system. O_2_-dependent degradation occurs after hydroxylation of HIFα by HIF-prolyl hydroxylase proteins PHD1, PHD2, and PHD3, and ubiquitination of hydroxylated HIFα by the von Hippel–Lindau disease tumor suppressor protein (VHL)-E3 ubiquitin ligase complex [13]. Reduced oxygen levels in the blood result in tissue hypoxia, which is the main stimulus for increased expression of EPO [14]. In hypoxia, the availability of oxygen for hydroxylation of HIFα by PHDs is limited. Consequently, HIFα protein is stabilized and translocated to the nucleus, where it forms a heterodimer with ARNT. The HIFα-ARNT complex binds to hypoxia-responsive elements (HREs) in the promoter/enhancer region of the *EPO* gene and recruits central transcriptional co-activators, histone acetyltransferase p300 (p300) and CREB-binding protein (CBP), through HIFα transactivation domain (TAD). This process is necessary for full transcriptional activation of the HIF complex [15]. 

Three independent HIFα subunits have been identified, namely HIF1α, HIF2α, and HIF3α. HIF1α and HIF2α, which are closely related and extensively studied, are both transcriptional activators with distinct as well as common gene targets, while the role of HIF3α in the regulation of gene transcription is less clear. It has been assumed that HIF3α plays an antagonistic role in the regulation of HIF1α and HIF2α isoforms [16]. Among the PHD and HIFα isoforms related to renal EPO production, PHD2 and HIF2α play the predominant role [17,18]. Among the ARNT isoforms, Aryl hydrocarbon receptor nuclear translocator-like protein 1 (ARNTL) has been identified as HIF2α partner, although it seems to be the most potent partner of HIF2α in chondrocytes. ARNT2 isoform, on the other hand, is more brain-specific and a dimerization partner for HIF1α in neural cells [19,20]. However, gene targeting studies have shown that neither ARNT2 nor ARNTL contribute to hypoxia-inducible transcriptional regulation [21].

The transcriptional activity of HIFα subunits can be negatively regulated also through another O_2_-dependent post-translational modification by asparaginyl hydroxylase Hypoxia-inducible factor 1-alpha inhibitor (FIH-1). FIH-1 mediates hydroxylation of HIFα TAD and thus prevents the binding of co-activators CBP/p300. However, FIH-1 more efficiently hydroxylates HIF1α than HIF2α [22]; therefore, PHD2 remains the primary oxygen sensor in controlling the EPO production [23].

The key players of well-established mechanisms in the oxygen-sensing pathway are presented in Table 1. Additional molecular mechanisms in the HIF-EPO pathway, which can modulate the stability or activity of HIF2α and its partners correlated with erythropoiesis, are presented in Figure 1 and included genes described further in details in this chapter. All described genes are reviewed at the end of the chapter in Table 2, including abbreviation list of genes and proteins.

### 2.1. Regulation of HIF2α Transcription

Very little information exists on how the expression of the HIF2α gene (*EPAS1*) in the kidney and liver is controlled. Moniz et al. (2015) [24] showed that the E2F1 transcription factor is required for expression of the HIF2α mRNA by directly binding to the HIF2α promoter in a diverse range of cell lines, including human kidney 786-O cells. They also demonstrated that the mRNA expression of HIF2α is regulated indirectly by the deubiquitylase Cezanne, which modulates the stability of E2F1. On the other hand, MCP-induced protein 1 has been reported to negatively regulate *EPAS1* transcription by acting as a regulator of stability and half-life of transcript encoding HIF2α in human renal cell carcinoma cell line [25]. However, several studies have investigated *EPAS1* transcriptional regulation in other types of cells. For example, in mouse adipocyte cells the involvement of transcription factors SP1 and SP3 in *EPAS1* expression has been suggested [26], while in neuroblastoma cells, phosphatidylinositol 3-kinase (PI3K) and serine/threonine-protein kinase mTORC2 have been demonstrated to regulate its transcription [27]. However, Hamidian et al. (2018) [28] identified a set of proteins specifically bound to the *EPAS1* promoter region at normoxia in neuroblastoma cells. The majority of these proteins showed decreased binding to the *EPAS1* promoter at hypoxia, indicating increased *EPAS1* transcription as the result of dissociated promoter-associated factors. A mechanism of negative feedback regulation for the *EPAS1* gene was also suggested in non-small cell lung cancer cells, where HIF2α mRNA was regulated by DNA methyltransferases (DNMTs). DNMT1 was verified as a HIF2α target gene which further promotes the hyper-methylation of *EPAS1* promoter and thus decreases mRNA expression levels [29].

### 2.2. Protein Interactions and Modifications of the Main Players in the Oxygen-Sensing Pathway

Apart from O_2_-dependent hydroxylation, HIFα proteins can be regulated through other post-translational modifications, such as phosphorylation, acetylation, methylation, ubiquitination, and sumoylation, and many do not require O_2_ as a co-factor. These modifications can regulate HIFα proteins directly or can occur on their binding partner, affecting HIFα activity or stability indirectly. Due to its important role in cancer development and progression, several studies in this regard have been directed to HIF1α (for review see [13]). However, we managed to identify quite a few HIF2α associated post-translational modifications as well as other proteins, interacting directly with HIF2α or its partners.

#### 2.2.1. Post-Translational Modifications

Phosphorylation of HIF2α has been shown to affect HIF2α transcriptional activity as well as its nuclear localization. Several kinases were found in this respect, such as casein kinases (CK) and mitogen-activated protein kinases (MAPK). For example, HIF2α transactivation capacity has been shown to depend on phosphorylation of HIF2α C-TAD at conserved threonine residues by CKIIα. This modification increased the affinity of HIF2α for CBP [15,30]. On the other hand, phosphorylation of HIF2α by MAPK1/3 enhanced HIF2α transcriptional activation by facilitating its nuclear accumulation via inhibition of its association with exportin-1 (CRM1) transporter [31]. Also, HIF2α modification by CKIδ led to its nuclear accumulation and full HIF2 transcriptional activity while silencing or inhibition of CKIδ reduced the expression of HIF2 target genes and the secretion of EPO in Huh7 and HepG2 hepatic cancer cell lines [32].

Acetylation of HIFα is a much less studied post-translational modification but can affect the transcriptional activity of modified proteins. Direct interaction and deacetylation of HIF2α with NAD-dependent protein deacetylase sirtuin-1 (hSIRT1) has been found to increase HIF2α transcriptional activity, affecting *EPO* gene expression. However, hSIRT1 has been demonstrated as a redox-sensing deacetylase [33]. Besides, hSIRT1 has been reported to mediate deacetylation of p300 that results in repression of p300 transcriptional activity in human embryonic kidney cells (HEK 293T). It has been suggested that this process requires SUMO modification of the p300 transcriptional repression domain (CRD1) [34].

Methylation has been mainly shown to affect DNA itself, regulating the transcription of genes. DNA methyltransferases that regulate HIF2α mRNA expression (described above) have been found also to affect the VHL gene expression. Even more, Yang et al. (2021) [35] proposed that VHL methylation contributes toward excessive erythropoiesis in chronic mountain sickness due to induced DNMT3a and DNMT3b expression triggered by hypoxia. Methylation at the CpG site in the VHL promoter resulted in decreased VHL expression and increased HIF2α and EPO expression in the bone marrow of rats.

Sumoylation is a highly dynamic post-translational modification that can be reversed by the action of SUMO-specific proteases. This process has been found to modify many proteins that participate in the oxygen-sensing pathway and EPO expression, including VHL, ARNT, p300, and CBP, resulting in repression of their transcriptional activity or inactivation, wherein SUMO conjugating enzyme UBC9 and E3 SUMO-protein ligases PIAS1 and PIAS4 apparently play major roles [36,37,38,39]. Van Hagen et al. (2009) [40] identified the SUMO acceptor site also in HIF2α and demonstrated that sumoylation enabled proteasomal degradation of the protein in human adenocarcinoma HeLa cells. On the other hand, Sentrin-specific proteases 1 and 3 modulate desumoylation of HIF2α and p300, respectively, resulting in protein stabilization or induction of its transcriptional activity [40,41].

Ubiquitination is another highly dynamic process, in the oxygen-sensing pathway best known for its role in VHL-HIFα proteasomal degradation. Apart from that, ubiquitin modification is more complex with diverse functions and can affect other proteins than HIFα [42]. VHL itself can be subjected to proteasomal degradation which consequently results in HIFα stability. Several factors that regulate VHL abundance and function have been identified, such as ubiquitin ligases Cadherin 1, WD repeat and SOCS box-containing protein 1 (WSB1), hSMURF1, and ubiquitin-conjugating enzyme E2-EPF [42,43,44,45]. Although WSB1 and E2-EPF expression is associated with cancer progression, they were both identified to interact with VHL and promote its ubiquitination and proteasomal degradation also in normoxic conditions in the human embryonic kidney or renal carcinoma cells [44,45]. In another cancer-directed study, Cadherin 1 has been shown to regulate VHL in a cell cycle-dependent manner in renal cell carcinoma cells [43]. SIAH1 has been reported as another ubiquitin ligase that can regulate the oxygen-sensing pathway by interacting with FIH-1 and facilitating its proteasomal degradation, which results in HIFα stability, although FIH-1 has a more prominent role in the hydroxylation of HIF1α than HIF2α [22]. Besides, ring finger protein 4 (RNF4) has been reported to act as a SUMO-targeted ubiquitin ligase for proteasomal degradation of sumo-modified HIF2α [40]. On the other hand, deubiquitinating enzymes (USPs) have been found to reverse the process of ubiquitination. For example, USP8 and USP29 can prevent VHL-mediated degradation of ubiquitinated HIF2α and thus increase its stability [46,47]. Probable ubiquitin carboxyl-terminal hydrolase FAF-X (USP9X) deubiquitylates hSMURF1, which targets VHL for proteasomal degradation, resulting in reduced VHL protein levels and stable HIFα protein [42].

#### 2.2.2. Other Protein Interactions

Besides post-translational modifications, the activity of HIFs can be modulated also by other protein interactions. We found protein interactions that can affect HIF2α directly or affect its partners in the well-established oxygen-sensing pathway. Many proteins were found to interact with PHD2 and VHL, and positively or negatively regulate HIF2α hydroxylation or ubiquitination and thus HIF2α stability. 

HIF2α modulators. NF-kappa-B essential modulator (NEMO) is a HIF2α interacting protein that seems to assist in recruiting CBP/p300 and enhance HIF2α transcriptional activity in human embryonic kidney cell line at normoxia [22]. Kelch-like protein 20 (KLHL20), as a part of an E3 ligase complex, interacts with HIF2α and increases its stability and activity. However, the molecular mechanism of the KLHL20-dependent regulation of HIF2α is not clear [48]. Another E3 ligase that binds to and increases HIF2α transcriptional activity instead of causing its degradation, as in the case of HIF1α, is a hypoxia-associated factor (HAF). HAF is overexpressed in a variety of tumors and has been associated with tumor development and poor prognosis [49]. On the other hand, Class E basic helix-loop-helix protein 41 (bHLHe41) has been shown to act as a tumor suppressor by binding to HIFα subunits and facilitating HIFα interaction with the proteasome and its degradation in an oxygen-, VHL-, and ubiquitination-independent manner [50]. bHLHe41 has also been found to be associated with erythrocytosis [51]. Chen et al. (2007) [52] reported the eukaryotic translation initiation factor 3 subunit E (eIF3e) to be a negative regulator of HIF2α stability. Specific direct interaction of HIF2α (but not HIF1α or HIF3α) with eIF3e led to HIF2α degradation via the proteasome pathway in a hypoxia- and VHL-independent manner in renal cell carcinoma 786-O cells.

PHD2 modulators. PHD2 is regulated by interactions with Peptidyl-prolyl cis-trans isomerase FKBP8 (PPIase FKBP8), Protein OS-9 (OS-9), Protein CBFA2T3 (CBFA2T3), and Prostaglandin E synthase 3 (PTGES3). While PPIase FKBP8 has been shown to decrease PHD2 protein stability and half-life [53], OS-9 promotion of PHD2 hydroxylase activity and consequent ubiquitination and proteasomal degradation of HIF(1)α has been reported in human embryonic kidney HEK293 cells. Also, CBFA2T3 interaction with PHD2 has been demonstrated to promote ubiquitination and proteasomal degradation of HIF(1)α in normoxia and hypoxia in human lymphoma cells. On the contrary to OS-9 which is overexpressed in osteosarcomas, CBFA2T3 is highly expressed in hematopoietic progenitor and erythroid cells [54,55]. The direct interaction between PHD2 and PTGES3 and its functional importance in promoting efficient HIFα degradation has been evidenced in the study of Song et al. (2013) [56], where PTGES3 knockdown augmented hypoxia-induced HIFα levels and HIF target genes. Since PTGES3 is known to bind to the N-terminus of HSP90, they proposed a model where PTGES3 recruits PHD2 to the HSP90 pathway, and the association between these three proteins facilitates hydroxylation of HIFα proteins. A protein which seems to be quite important to mention is LIM domain-containing protein 1 (LIMD1). PHD2 binding interface responsible for interaction with LIMD1 has been suggested in bioinformatics investigation of erythrocytosis causative mutations in PHD2 [57]. Actually, LIMD1 was shown to form a complex with PHD2 and VHL in several cell types, including human embryonic kidney cells, probably enhancing the activity of both proteins and yielding efficient HIF(1)α proteasomal degradation [58]. 

VHL modulators. VHL is the substrate recognition component of Cullin Ring Ubiquitin ligase complex (CRL), composed of Elongins B and C (ELOB and ELOC), scaffold protein Cullin-2, and ubiquitin-protein ligase RBX1. These components all play an important role in VHL-dependent ubiquitination and degradation of HIF2α [59]. ELOB and ELOC are complex adaptor subunits and important for the stabilization of VHL itself, while the interaction between VHL and ELOC can be stabilized by Thialysine N-epsilon-acetyltransferase (SSAT-2). T-complex protein 1 subunit alpha (TRiC), on the other hand, facilitates VHL folding, which enables its interaction with ELOB and ELOC to form the CRL complex. SSAT-2 and TRiC both seem important in the regulation of HIFα proteasomal degradation since failure to generate a correctly folded VHL or a mature CRL complex results in degradation of VHL and increased stability of HIFα subunits [60,61]. Contrary to above mentioned VHL interaction proteins, DNA-binding protein inhibitor ID-2 has been reported to be responsible for disrupting the interaction of VHL and HIF2α and increasing the stability of HIF2α in an O_2_-dependent manner, although ID2 is an oncoprotein and the study was performed in glioblastoma cells [62]. 

FIH-1 modulators. Several proteins have been reported also to interact with FIH-1, such as NF-kappa-B inhibitor alpha, Nuclear factor NF-kappa-B p105 subunit, Neurogenic locus notch homolog proteins 1, 2, and 3, and Ankyrin repeat and SOCS box protein 4. It has been suggested that these proteins are substrates for hydroxylation mediated by FIH-1 which can take place in competition with HIFα hydroxylation, which could potentially regulate HIF transcriptional activity [22]. Besides, Homeobox protein cut-like 1 has been found to be an important repressor of FIH-1 transcription that can contribute to increased HIFα stability and transcriptional activity. However, its binding to the FIH-1 promoter is supposed to be regulated by Protein kinase C zeta type [63].

ARNT modulators. Aryl hydrocarbon receptor (AHR), which plays a prominent role in xenobiotic metabolism, has attracted our attention since ARNT can be its dimerization partner. Crosstalk between the AHR and hypoxia pathways was evident in a study, where endogenous AHR ligand suppressed nuclear accumulation of HIF2α and subsequent EPO production in human hepatoma HepG2 cells [64]. A noteworthy ARNT interacting protein, which was identified in human embryonic kidney 293T cells, is EPO-inducible transforming acidic coiled-coil-containing protein 3 (TACC3). This protein can act as a co-activator of the HIF2 complex, and it has been demonstrated that the modular ARNT PAS-B domain simultaneously engages its heterodimeric HIFα partner and TACC3 [65].

#### 2.2.3. Nuclear Transport of the Main Players in the Oxygen-Sensing Pathway

Proteins that are larger than ∼40 kDa move between the nucleus and cytoplasm through the nuclear pore complex (NPC). This process is regulated by nuclear transport receptors importins and exportins that recognize specific transport signals, the nuclear localization signal (NLS) and the nuclear export signal (NES), respectively. Importinα serves as an adapter protein that recognizes and binds directly to proteins containing either a simple or bipartite NLS and forms a trimeric import cargo complex with importinβ, which mediates the interaction with the NPC. Six human importinα isoforms, which differ in affinities to their targets, have been characterized. However, most proteins can directly bind the importinβ and do not utilize importinα as an adaptor protein. NES are recognized by nuclear export receptors, with CRM1 being the most abundant among seven exportins described so far [66,67].

The importance of intracellular transport of the oxygen-sensing machinery has been relatively ignored. However, the regulation of translocation between cytoplasm and nucleus represents another level of activity control of HIF subunits and molecular basis for cellular oxygen homeostasis. A classical importin α/β-dependent NLS has been found in HIF2α as well as in ARNT, and both have been shown to bind to importins α1, α3, α5 and α7 isoforms in HeLa and U2OS cells. HIF2α can be targeted also solely by importinβ, even with higher efficiency than by importinα [68]. While HIFα subunits shuttle between nucleus and cytoplasm, ARNT is permanently inside the nucleus [66]. The nuclear export of HIFα is mediated through NES that are recognized by CRM1. However, HIF2α has been shown to associate with CRM1 in a phosphorylation-sensitive manner that is controlled by MAPK1/3 [31]. 

Transport between nucleus and cytoplasm has been reported also for PHD2 and VHL. Although the nuclear import of PHD2 does not occur via classical importin α/β interaction, its nuclear export depends on CRM1 binding [66]. Nuclear export of VHL, on the other hand, is regulated through the transcription-dependent nuclear export motif (TD-NEM), and Elongation factor 1-alpha 1 is involved in TD-NEM-mediated nuclear export of this protein [69]. Most of the VHL protein is found in the cytosol under steady-state conditions. 

### 2.3. Regulation of EPO Transcription

As already mentioned, the major site of EPO expression is the kidney, while the liver is the main source of EPO during fetal development. The adult liver maintains the potential for high expression of *EPO* mRNA but production at this site is physiologically repressed. Transgenic animal studies revealed that HRE in renal cells is located between 9.5 and 14 kb 5′ of *EPO*, while HRE in the liver lies within 0.7 kb 3′ of the gene. A specific negative regulatory liver element is located between 2.2 and 7 kb 3′ to the gene, and generally active repressive elements exist between 0.4 and 6 kb 5′. Due to the lack of an appropriate renal cell culture system, the mechanisms of hypoxia-induced *EPO* expression were analyzed with the help of human hepatoma cells. Hypoxia-inducible enhancer has been identified at 120 bp 3′ to the *EPO* polyadenylation site, containing three important hypoxia-responsive sites: a CACGTGCT sequence at the 5′-end, defined as a HIF binding site (HBS); a CACA repeat, located 7 bp further 3′ of the HBS of the *EPO* gene; and a direct repeat of two steroid receptor half-sites separated by two base pairs (DR-2 site) at 3′ *EPO* enhancer [14,70]. *EPO* promoter, located at the 5′ flanking region of the gene, has been reported as a weak promoter that still contributes to the hypoxic inducibility of the *EPO* gene and synergistically cooperates with an enhancer [71]. The minimal *EPO* promoter capable of induction by hypoxia encompasses 117 bp 5′ to the transcription initiation site but lacks the consensus HBS and is negatively regulated by GATA sequence [14,70].

Besides HIF2 complex and transcriptional co-activators p300 and CBP, *EPO* expression depends on the cooperation with other transcription factors that associate with the *EPO* gene, such as Hepatocyte nuclear factor-4α (HNF4α). HNF4α binds to the DR-2 site in the *EPO* enhancer and probably plays a critical role in activating hepatic and renal hypoxia-inducible *EPO* transcription [72]. However, Makita et al. (2001) [73] demonstrated that Retinoid X receptor alpha (RXRα) is involved in transcriptional regulation of the *EPO* gene in the fetal liver during mouse development and also binds to the DR-2 site of *EPO* enhancer. Furthermore, nuclear receptor coactivator 3 (NCoA-3), a histone acetyltransferase coactivator, is recruited to the 3′ enhancer of the *EPO* gene upon hypoxic stimulation in a p300-dependent manner and NCoA-3 knockdown reduced *EPO* transcriptional activity in hepatoma Hep3B cells [71]. *EPO* gene contains also NFKB3 sequences, and NFKB3 has been reported to play a key role in hypoxia-regulated *EPO* gene expression [74,75].

On the other hand, transcription factors GATA-2 and GATA-3 have been shown to negatively regulate the *EPO* transcription through GATA sites in *EPO* promoter in hepatic and distal tubular cells [76,77]. Another mechanism repressing *EPO* expression has been shown in the study of Yin and Blachard (2000) [78], where CpG islands in the promoter and 5′-untranslated region (5′-UTR) of the *EPO* gene have been identified. The authors proposed that methylation of CpGs in the promoter represses *EPO* transcription by blocking the binding of sequence-specific DNA binding proteins, while methylation in the 5′-UTR recruits Methyl-CpG binding protein 2, and directly represses transcription or recruits co-repressors, histone deacetylases, or both.

**Table 2 ijms-22-07074-t002:** Genes and molecular mechanisms correlated with erythropoiesis in the HIF-EPO pathway.

Gene Symbol	Gene ID	Protein Name	Category	Mechanism	Ref.
**Regulation of HIF2α transcription**					
*E2F1* ^#^	1869	Transcription factor E2F1	1	Transcriptional activation of *EPAS1*	[24]
*OTUD7B* ^#^	56957	OTU domain-containing protein 7B (Cezanne)	2	Transcriptional activation of *EPAS1*	[24]
*ZC3H12A* ^#^	80149	MCP-induced protein 1 (MCPIP1)	1	Negative transcriptional regulation of *EPAS1*	[25]
*SP1* ^#^	6667	Transcription factor Sp1	1	Transcriptional activation of *EPAS1*	[26]
*SP3* ^#^	6670	Transcription factor Sp3	1	Transcriptional activation of *EPAS1*	[26]
*PI3K* ^#^		Phosphatidylinositol 3-kinase	2	Transcriptional activation of *EPAS1*	[27]
*MTOR* ^#^	2475	Serine/threonine-protein kinase mTOR	2	Transcriptional activation of *EPAS1*	[27]
*DNMT1* ^#^	1786	DNA (cytosine-5)-methyltransferase 1 (DNMT1)	1	Negative transcriptional regulation of *EPAS1*	[29]
**HIF2α protein modifications and protein interactions**					
*HIF1AN*	55662	Hypoxia-inducible factor 1-alpha inhibitor (FIH-1)	1	Hydroxylation	[22]
*CSNK2A1*	1457	Casein kinase II subunit alpha (CKIIα)	1	Phosphorylation	[30]
*CSNK1D* ^#^	1453	Casein kinase I isoform delta (CKIδ)	1	Phosphorylation and nuclear transport	[32]
*MAPK 1/3* ^#^	5594/5595	Mitogen-activated protein kinase 1/3 (MAPK 1/3)	1	Phosphorylation and nuclear transport	[31]
*SIRT1* ^#^	23411	NAD-dependent protein deacetylase sirtuin-1 (hSIRT1)	1	Deacetylation	[33]
*SENP1* ^#^	29843	Sentrin-specific protease 1	1	Desumoylation	[40]
*RNF4* ^#^	6047	E3 ubiquitin-protein ligase RNF4	1	Ubiquitination	[40]
*USP8* ^#^	9101	Ubiquitin carboxyl-terminal hydrolase 8 (USP8)	1	Deubiquitination	[46]
*USP29*	57663	Ubiquitin carboxyl-terminal hydrolase 29 (USP29)	1	Deubiquitination	[47]
*IKBKG*	8517	NF-kappa-B essential modulator (NEMO)	1	Other interaction proteins	[22]
*KLHL20* ^#^	27252	Kelch-like protein 20 (KLHL20)	1	Other interaction proteins	[48]
*SART1 (HAF)*	9092	Hypoxia-associated factor (HAF)	1	Other interaction proteins	[49]
*BHLHE41 (SHARP1)*	79365	Class E basic helix-loop-helix protein 41 (bHLHe41)	4	Other interaction proteins	[50]
*EIF3E (INT6)*	3646	Eukaryotic translation initiation factor 3 subunit E (eIF3e)	1	Other interaction proteins	[52]
**PHD2 protein modifications and protein interactions**					
*FKBP8*	23770	Peptidyl-prolyl cis-trans isomerase FKBP8 (PPIase FKBP8)	1	Other interaction proteins	[53]
*OS-9*	10956	Protein OS-9 (OS-9)		Other interaction proteins	[54]
*CBFA2T3 (MTG16)* ^#^	863	Protein CBFA2T3 (CBFA2T3)	1	Other interaction proteins	[55]
*PTGES3 (P23)*	10728	Prostaglandin E synthase 3 (PTGES3)	1	Other interaction proteins	[55]
*LIMD1*	8994	LIM domain-containing protein 1	1, 4	Other interaction proteins	[57,58]
**VHL protein modifications and protein interactions**					
*LIMD1*	8994	LIM domain-containing protein 1	1, 4	Other interaction proteins	[58]
*DNMT3A*	1788	DNA (cytosine-5)-methyltransferase 3A (DNMT3a)	1	MethylationNegative transcriptional regulation of *VHL*	[35]
*DNMT3B*	1789	DNA (cytosine-5)-methyltransferase 3B (DNMT3b)	1	MethylationNegative transcriptional regulation of *VHL*	[35]
*UBE2S*	27338	Ubiquitin-conjugating enzyme E2 S (E2-EPF)	1, 4	UbiquitinationDegradation of VHL	[45]
*PIAS4*	51588	E3 SUMO-protein ligase PIAS4 (PIASY)	1, 4	Sumoylation	[39]
*CDH1*	999	Cadherin 1	1	UbiquitylationDegradation of VHL	[43]
*WSB1*	26118	WD repeat and SOCS box-containing protein 1 (WSB1)	1	UbiquitinationDegradation of VHL	[44]
*SMURF1*	57154	E3 ubiquitin-protein ligase SMURF1 (hSMURF1)	1	UbiquitinationDegradation of VHL	[42]
*USP9X*	8239	Probable ubiquitin carboxyl-terminal hydrolase FAF-X	2, 4	DeubiquitylationDegradation of VHL	[42]
*ELOB*	6923	Elongin-B (ELOB)	2	Other interaction proteinsUbiquitination of HIFα	[59]
*ELOC*	6921	Elongin-C (ELOC)	1	Other interaction proteinsUbiquitination of HIFα	[59]
*CUL2*	8453	Cullin-2 (CUL-2)	2	Other interaction proteinsUbiquitination of HIFα	[59]
*RBX1*	9978	E3 ubiquitin-protein ligase RBX1	2	Other interaction proteinsUbiquitination of HIFα	[59]
*SAT2*	112483	Thialysine N-epsilon-acetyltransferase (SSAT-2)	1, 4	Other interaction proteins	[60]
*TCP1*	6950	T-complex protein 1 subunit alpha (TRiC)	1	Other interaction proteins	[61]
*ID2* ^#^	3398	DNA-binding protein inhibitor ID-2 (ID2)	1	Other interaction proteins	[62]
**FIH protein modifications and protein interactions**					
*SIAH1*	6477	E3 ubiquitin-protein ligase SIAH1	2	Ubiquitination	[22]
*NFKBIA*	4792	NF-kappa-B inhibitor alpha	2	Other interaction proteins	[22]
*NFKB1*	4790	Nuclear factor NF-kappa-B p105 subunit	2	Other interaction proteins	[22]
*NOTCH1/2/3*	4851/4853/4854	Neurogenic locus notch homolog protein 1/2/3 (Notch 1/2/3)	2	Other interaction proteins	[22]
*ASB4*	51666	Ankyrin repeat and SOCS box protein 4 (ASB-4)	2	Other interaction proteins	[22]
*CUX1*	1523	Homeobox protein cut-like 1	2	DNA protein partner	[63]
*PRKCZ*	5590	Protein kinase C zeta type	3	Other interaction proteins	[63]
**ARNT protein modifications and protein interactions**					
*UBE2I* ^#^	7329	SUMO-conjugating enzyme UBC9	2	Sumoylation	[36]
*PIAS1* ^#^	8554	E3 SUMO-protein ligase PIAS1	2	Sumoylation	[36]
*AHR* ^#^	196	Aryl hydrocarbon receptor (AHR)	2	DNA protein partner	[64]
*TACC3*	10460	Transforming acidic coiled-coil-containing protein 3	2	DNA protein partnerCo-activator of HIF2 complex	[65]
**Nuclear transport of key players of oxygen-sensing pathway**					
*KPNA2* ^#^	3838	Importin subunit alpha-1 (Importinα)	1	Nuclear import of HIF2α, ARNT	[68]
*KPNB1* ^#^	3837	Importin subunit beta-1 (Importinβ)	1	Nuclear transport of HIF2α, ARNT	[68]
*XPO1*	7514	Exportin-1 (CRM1)	1	Nuclear export of HIF2α, PHD2	[31,66]
*EEF1A1*	1915	Elongation factor 1-alpha 1 (eEF1A)	1	Nuclear export of VHL	[69]
**Regulation of EPO transcription**					
*EP300*	2033	Histone acetyltransferase p300 (p300)	1	Transcriptional activation of EPO (co-activator)	[15,71]
*CREBBP*	1387	CREB-binding protein (CBP)	1	Transcriptional activation of EPO (co-activator)	[15]
*HNF4A*	3172	Hepatocyte nuclear factor-4α (HNF4α)	1	Transcriptional activation of EPO	[72]
*RXRA*	6256	Retinoid X receptor alpha (RXRα)	1	Transcriptional activation of EPO	[73]
*NCOA3 (SRC3)* ^#^	8202	Nuclear receptor coactivator 3 (NCoA-3)	1, 4	DNA protein partner	[71]
*RELA*	5970	Transcription factor p65 (NFKB3)	1, 4	Transcriptional activation of *EPO*	[74,75]
*GATA2*	2624	Endothelial transcription factor GATA-2 (GATA-2)	1	Negative transcriptional regulation of EPO	[77]
*GATA3*	2625	Trans-acting T-cell-specific transcription factor GATA-3 (GATA-3)	1	Negative transcriptional regulation of EPO	[77]
*MECP2*	4204	Methyl- CpG binding protein 2 (MeCP2)	1	Negative transcriptional regulation of EPO	[78]
**p300 protein modifications and protein interactions**					
*SIRT1*	23411	NAD-dependent protein deacetylase sirtuin-1 (hSIRT1)	2	Deaceltylation	[34]
*UBE2I*	7329	SUMO-conjugating enzyme UBC9	2	Sumoylation	[37]
*SENP3*	26168	Sentrin-specific protease 3	2	Desumoylation	[41]
**CBP protein modifications and protein interactions**					
*UBE2I*	7329	SUMO-conjugating enzyme UBC9	2	Sumoylation	[38]

^#^ Experiments were performed in cancer cell lines, or cells other than renal cells, or it was a cancer-directed study. Category: 0—genes involved in erythropoiesis; 1—protein partners of 0; 2—protein partners of 1; 3—protein partners of 2; 4—protein partners of 0 or 1, but only proven with HIF1α or cancer study. Abbreviation list: CBFA2T3—core-binding factor, runt domain, alpha subunit 2; translocated to, 3; CREB—cAMP-responsive element-binding protein; CRM1—chromosomal region maintenance 1; E2F—E2 factor; EP300—E1A binding protein p300; FAF-X—Fat facets protein-related, X-linked; FIH—Factor Inhibiting HIF; FKBP8—FK506 binding protein 8; ID—inhibitor of DNA binding; IKBKG—inhibitor of nuclear factor kappa B kinase regulatory subunit gamma; KPNA—karyopherin subunit alpha; KPNB—karyopherin subunit beta; MCP- monocyte chemotactic protein; MTG—myeloid translocation gene; mTOR—mechanistic target of rapamycin kinase; OS-9—Osteosarcoma Amplified 9; PIAS—protein inhibitor of activated STAT; RBX1—ring box 1; RNF—RING finger; SART1—spliceosome associated factor 1, recruiter of U4/U6.U5 tri-snRNP; SAT2/SSAT-2—spermidine/spermine N1-acetyltransferase family member 2; SHARP1—Enhancer-of-split and hairy-related protein 1; SIAH—Seven In Absentia Homolog; SMURF1—Smad ubiquitination regulatory factor 1; SOCS—suppressor of cytokine signalling; Sp—specificity protein; SRC3—Steroid receptor coactivator protein 3; SUMO—small ubiquitin-like modifier; TRiC—TCP1 ring complex; UBC9—ubiquitin carrier protein 9; UBE2I—ubiquitin conjugating enzyme E2 I; USP—ubiquitin specific peptidase; ZC3H12A—zinc finger CCCH-type containing 12A.

## 3. Diseases and Putative Treatments

Described factors may be involved in deregulation of erythropoiesis, including anemia, erythrocytosis and cancer [8]. Anemia is a deficiency of mature RBCs, which leads to a decreased oxygen-carrying capacity of the blood, tissue hypoxia, and a variety of clinical consequences, and is a very common disorder [79]. A severe anemia resulting from a mutation in *EPO* has been described in a study by Kim et al. (2017) [80]. Erythrocytosis, on the other hand, is a rare hematological disorder with increased RBC count, Hb concentration, and hematocrit (Hct) above the reference range. Four types of erythrocytosis are known to be caused by mutations in genes encoding key players of oxygen-sensing signalling and EPO production, namely *VHL* (ECYT2), *EGLN1* (ECYT3), *EPAS1* (ECYT4) and *EPO* (ECYT5) [81,82]. Importantly, HIF1α, HIF2α, VHL and EPO also play a well-known role in development and progression of a variety of human cancers [8,83,84,85,86]. Therefore, the HIF-EPO pathway is a potential target for treatment of cancer. 

Also disorders affecting high-altitude accommodation can be mediated by HIF-EPO pathway deregulation. Tight regulation of EPO production is responsible for removal of erythrocytes and normalization of increased hemoglobin (Hb) upon descent, which are typical observations of acclimatization in long-term sojourners at high altitude and in high-altitude residents [87]. Although it has been assumed that this decrease occurs through neocytolysis, the selective destruction of newly formed erythrocytes during stress-erythropoiesis in hypoxia [88], Klein et al. (2021) [87] confirmed that the decrease actually occurs by a reduced rate of erythropoiesis along with normal clearance of senescent erythrocytes.

Several HIF inhibitors have been developed and investigated up to now, comprising direct inhibitors which affect the expression or function of HIF proteins through inhibition of mRNA expression, protein synthesis, dimerization of α and β subunits, DNA binding, and transcriptional activities of HIF, and indirect inhibitors which regulate proteins of upstream or downstream pathways and thus affect the HIF signaling, such as mTOR inhibitors [86,89,90]. Histone deacetylase inhibitors are also explored as anti-cancer agents, since they repress HIF1α and HIF2α transactivation potential [91]. Additionally, a promising potential of HIF inhibitors has been reported for treatment of erythrocytosis [92] and anemia [93]. Besides, recombinant human EPO and other erythropoiesis stimulating agents are already widely used to treat anemias associated with low EPO levels [79] as well as in cancer therapy [94], and GATA inhibitors have been shown to restore erythropoiesis in a mouse model of anemia of chronic disease [95]. 

## 4. Conclusions

Mechanisms controlling the HIF-EPO pathway have a prominent role in regulation of EPO synthesis, that are all together affecting the RBC production. Overall, any imbalance in HIF-EPO pathway mechanisms can result in deregulation of erythropoiesis that can lead to several diseases, including anemia, erythrocytosis and cancer. Understanding the main regulatory pathways in the process of erythropoiesis is necessary to identify the causative factors which can contribute to the development of such pathological conditions, and help to improve the diagnostic approach and therapeutic implications. Apart from the well-known O_2_-dependent hydroxylation mechanism that coordinates HIF2α stability and transcriptional activity and EPO synthesis, many other proteins are contributing to post-translational modifications, nuclear transport, and transcriptional regulation of complete oxygen-sensing machinery but have been relatively ignored in this context. It should be noted that besides HIF1α, HIF2α and VHL also play a role in cancer development and progression. As the majority of available information and literature data are directed in cancer studies, their enrolment in the regulatory mechanisms in healthy tissue must be considered with caution. However, further analysis of mechanisms in EPO receptor signaling, Hb oxygen affinity pathway, and iron metabolism should also be investigated in this regard.

## Figures and Tables

**Figure 1 ijms-22-07074-f001:**
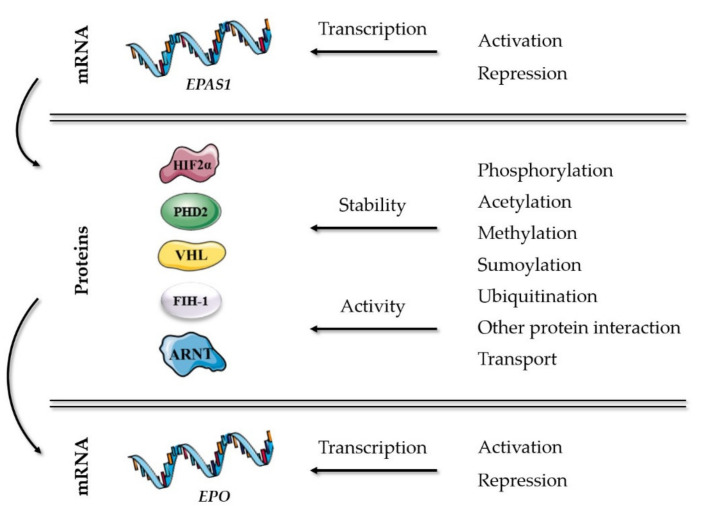
Molecular mechanisms in the HIF-EPO pathway, which can modulate the stability or activity of HIF2α and its partners correlated with erythropoiesis.

**Table 1 ijms-22-07074-t001:** Key players of oxygen-sensing pathway and EPO expression.

Gene Symbol	Gene ID	Protein Name	Mechanism
*EPAS1*	2034	Hypoxia-inducible factor 2-alpha (HIF2α)	Transcriptional activation of EPO [17]
*HIF1A*	3091	Hypoxia-inducible factor 1-alpha (HIF1α)	Transcriptional activation [16]
*HIF3A*	64344	Hypoxia-inducible factor 3-alpha (HIF3α)	Transcriptional repression [16]
*EGLN1*	54583	HIF-prolyl hydroxylase 2 (PHD2)	HIF2α hydroxylation [18]
*VHL*	7428	von Hippel-Lindau disease tumor suppressor protein (VHL)	HIF2α ubiquitination [16]
*ARNT*	405	Aryl hydrocarbon receptor nuclear translocator (ARNT)	HIF2α DNA protein partner Transcriptional activation of EPO [15]
*ARNTL* ^#^	406	Aryl hydrocarbon receptor nuclear translocator-like protein 1 (ARNTL)	HIFα DNA protein partner [19]
*ARNT2* ^#^	9915	Aryl hydrocarbon receptor nuclear translocator 2 (ARNT2)	HIFα DNA protein partner [20]
*EPO*	2056	Erythropoietin	

^#^ Experiments were performed in cancer cell lines, or cells other than renal cells, or it was a cancer-directed study.

## Data Availability

Not applicable.

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
