# Peer review of "Molecular Insights into the Oxygen-Sensing Pathway and Erythropoietin Expression Regulation in Erythropoiesis"

_ijms, 2021, doi:10.3390/ijms22137074_

Round 1
Reviewer 1 Report
As far as I can judge, Tome and Debeljak present a comprehensive review with the exception of one aspect, which definitely should go into a review dealing with oxygen-sensing, EPO and erythropoiesis and this is the topic of neocytolysis. This holds especially true since latest research (DOI: 10.1111/apha.1364) gave evidence that the reduction in total haemoglobin (when necessary) is not due to destruction of neocytes but just a result of EPO-mediated erythropoiesis regulation. This aspect (especially since it is latest state of knowledge) should be added to the review.
Author Response
GENERAL:
We thank all reviewers for detailed revision of the paper and indicated corrections, that improved our revision. All corrections suggested by reviewers including additional minor corrections by authors are marked with traced changes.
The abbreviation list of genes and protein is included in Table 2.
New references addressing tools, diseases, treatments and neocytolysis were including and reference list updated accordingly.
Reviewer 1
Comments and Suggestions for Authors
As far as I can judge, Tome and Debeljak present a comprehensive review with the exception of one aspect, which definitely should go into a review dealing with oxygen-sensing, EPO and erythropoiesis and this is the topic of neocytolysis. This holds especially true since latest research (DOI: 10.1111/apha.1364) gave evidence that the reduction in total haemoglobin (when necessary) is not due to destruction of neocytes but just a result of EPO-mediated erythropoiesis regulation. This aspect (especially since it is latest state of knowledge) should be added to the review.
As suggested by reviewer 1, we included the role of EPO in erythropoesis downregulation and reduction in haemoglobin, especially addressing neocytolysis. The paragraph is included in disease section (p. 13, line 778 – 784):
»Tight regulation of EPO production is also responsible for removal of erythrocytes and normalization of increased hemoglobin (Hb) upon descent, which are typical observations of acclimatization in long-term so-journers at high altitude and in high-altitude residents (Klein et al. 2021). Although it has been assumed that this decrease occurs through neocytolysis, the selective destruction of newly formed erythrocytes during stress-erythropoiesis in hypoxia (Mairbäurl 2018), Klein et al. (2021) (Klein et al. 2021) confirmed that the decrease actually occurs by a reduced rate of erythropoiesis along with normal clearance of senescent erythrocytes.«
Reviewer 2 Report
This is an interesting and well written review paper which outlines the Molecular pathways involved in the oxygen sensing pathway and erythropoietin expression regulation in erythropoiesis.
I only have a few comments for the authors, please see below:
- The abstract would benefit from some details of the actual gene and molecular contributors involved in the regulation oxygen sensing and erythropoietin expression in erythropoiesis together with the actual causative factors that may contribute to the development of haematological diseases.
- There was no clear aims for the review outlined after the introduction.
- This review would really benefit from some mechanistic diagrams that clearly outline the findings of the paper.
- A separate section outlining diseases and putative treatments for impaired regulation of erythropoiesis together with more detail on the connection with cancer.
- I would have liked more detail/explanation of the relationship under normal conditions/disease state between erythropoietin expression and iron metabolism. Is there a potential oxidative stress element ?
- A list of abbreviations would really help the reader.
Author Response
GENERAL:
We thank all reviewers for detailed revision of the paper and indicated corrections, that improved our paper. All corrections suggested by reviewers including additional minor corrections by authors are marked with traced changes.
The abbreviation list of genes and protein is included in Table 2.
New references addressing tools, diseases, treatments and neocytolysis were including and reference list updated accordingly.
Reviewer 2
Comments and Suggestions for Authors
This is an interesting and well written review paper which outlines the Molecular pathways involved in the oxygen sensing pathway and erythropoietin expression regulation in erythropoiesis.
I only have a few comments for the authors, please see below:
- The abstract would benefit from some details of the actual gene and molecular contributors involved in the regulation oxygen sensing and erythropoietin expression in erythropoiesis together with the actual causative factors that may contribute to the development of haematological diseases.
We do agree with the comment suggested by reviewer 2, and we added the details of the actual gene contributors involved in the regulation oxygen sensing and erythropoietin expression in erythropoiesis together with the actual causative factors that may contribute to the development of erythrocytosis in the abstract (p. 1, line 14 – 17 and 19 - 23):
“For example, mutations in genes encoding key players of oxygen-sensing pathway and regulation of EPO production (HIF-EPO pathway), namely VHL, EGLN, EPAS1 and EPO are well known causative factors that contribute to the development of erythrocytosis.«
»We identified genes encoding transcription factors and proteins that control transcriptional activation or repression; genes encoding kinases, deacetylases, methyltransferases, conjugating enzymes, protein ligases, and proteases involved in post-translational modifications; and genes encoding nuclear transport receptors that regulate nuclear transport.«
- There was no clear aims for the review outlined after the introduction.
We described the aim of the study more in detail at the end of Introduction section (p. 2, line 77 – 83):
»The aim of this study was to investigate the molecular mechanisms involved in the oxygen-sensing pathway and regulation of erythropoietin production that correlate with erythropoiesis. We reviewed factors acting on the level of (i) transcriptional regulation and (ii) protein interactions, including post-translational modifications and nuclear transport. All these factors alone or cumulatively modulate the stability or activity of the main players in the HIF-EPO pathway, and thus influence the EPO synthesis and final RBC production.«
- This review would really benefit from some mechanistic diagrams that clearly outline the findings of the paper.
We absolutely agree with this suggestion and we added a diagram outlining the findings of the paper at the beginning of the paper (p. 4, line 228 – 231).
- A separate section outlining diseases and putative treatments for impaired regulation of erythropoiesis together with more detail on the connection with cancer.
We added a separate section outlining diseases and putative treatments for impaired regulation of erythropoiesis in the separate section of the paper Diseases and putative treatments (p. 13, line 765 – 798):
»Described factors may be involved in deregulation of erythropoiesis, including ane-mia, erythrocytosis and cancer [8]. Anemia is a deficiency of mature RBCs, which leads to a decreased oxygen-carrying capacity of the blood, tissue hypoxia, and a variety of clinical consequences, and is a very common disorder [79]. A severe anemia resulting from a mu-tation in EPO has been described in a study of Kim et al. (2017) [80]. Erythrocytosis, on the other hand, is rare haematological disorder with increased RBC count, Hb concentration, and hematocrit (Hct) above the reference range. Four types of erythrocytosis are known to be caused by mutations in genes encoding key players of oxygen-sensing signalling and EPO production, namely VHL (ECYT2), EGLN1 (ECYT3), EPAS1 (ECYT4) and EPO (EC-YT5) [81, 82]. Importantly, HIF1α, HIF2α, VHL and EPO also play a well known role in development and progression of a variety of human cancers [8, 83-86]. Therefore, the HIF-EPO pathway is a potential target for treatment of cancer.
Tight regulation of EPO production is also responsible for removal of erythrocytes and normalization of increased hemoglobin (Hb) upon descent, which are typical obser-vations of acclimatization in long-term so-journers at high altitude and in high-altitude residents [87]. Disorders affecting high-altitude accmodation can also be mediated by HIF-EPO pathway deregulation. Although it has been assumed that this decrease occurs through neocytolysis, the selective destruction of newly formed erythrocytes during stress-erythropoiesis in hypoxia [88], Klein et al. (2021) [87] confirmed that the decrease actually occurs by a reduced rate of erythropoiesis along with normal clearance of senes-cent erythrocytes.
Several HIF inhibitors have been developed and investigated up to now, comprising of direct inhibitors which affect the expression or function of HIF proteins through inhibi-tion of mRNA expression, protein synthesis, dimerization of α and β subunits, DNA binding, and transcriptional activities of HIF, and indirect inhibitors which regulate pro-teins of upstream or downstream pathways and thus affect the HIF signaling, such as mTOR inhibitors [86, 89-90]. Histone deacetylase inhibitors are also explored as an-ti-cancer agents, since they repress HIF1α and HIF2α transactivation potential [91]. Addi-tionally, a promising potential of HIF inhibitors has been reported for treatment of eryth-rocytosis [92] and anemia [93]. Besides, recombinant human EPO and other erythropoie-sis stimulating agents are already widely used to treat anemias associated with low EPO levels [79] as well as in cancer therapy [94], and GATA inhibitors have been shown to re-store erythropoiesis in a mouse model of anemia of chronic disease [95].«
- I would have liked more detail/explanation of the relationship under normal conditions/disease state between erythropoietin expression and iron metabolism. Is there a potential oxidative stress element?
The aim of this review was to analyze the molecular mechanisms from oxygen-sensing pathway up to EPO expression regulation. We do agree, that the relationship between EPO synthesis and iron metabolism is also very important. However, we believe that this aspect is so complex that should be addressed in a separate paper.
- A list of abbreviations would really help the reader.
We do agree with the comment that the reader will follow the paper more easily with abbreviation list, therefore we included the list of genes and proteins in Table 2 (p. 13, line 753 – 764):
“Abbreviation list: CBFA2T3 - core-binding factor, runt domain, alpha subunit 2; translocated to, 3; CREB – cAMP-responsive element-binding protein; CRM1 - chromosomal region maintenance 1; E2F – E2 factor; EP300 - E1A binding protein p300; FAF-X - Fat facets protein-related, X-linked; FIH - Factor Inhibiting HIF; FKBP8 - FK506 binding protein 8; ID - inhibitor of DNA binding; IKBKG - inhibitor of nuclear factor kappa B kinase regulatory subunit gamma; KPNA - karyopherin subunit alpha; KPNB - karyopherin subunit beta; MCP- monocyte chemotactic protein; MTG - myeloid translocation gene; mTOR - mechanistic target of rapamycin kinase; OS-9 - Osteosarcoma Amplified 9; PIAS - protein inhibitor of activated STAT; RBX1 – ring box 1; RNF – RING finger; SART1 - spliceosome associated factor 1, recruiter of U4/U6.U5 tri-snRNP; SAT2/SSAT-2 - spermidine/spermine N1-acetyltransferase family member 2; SHARP1 - Enhancer-of-split and hairy-related protein 1; SIAH - Seven In Absentia Homolog; SMURF1 - Smad ubiquitination regulatory factor 1; SOCS – suppressor of cytokine signalling; Sp - specificity protein; SRC3 - Steroid receptor coactivator protein 3; SUMO - small ubiquitin-like modifier; TRiC - TCP1 ring complex; UBC9 - ubiquitin carrier protein 9; UBE2I - ubiquitin conjugating enzyme E2 I; USP - ubiquitin specific peptidase; ZC3H12A - zinc finger CCCH-type containing 12A.”
Reviewer 3 Report
In the manuscript entitled “Molecular insights into the oxygen-sensing pathway and erythropoietin expression regulation in erythropoiesis” the authors investigate the molecular mechanisms involved in the oxygen-sensing pathway and regulation of erythropoietin production (HIF-EPO pathway) that correlate with erythropoiesis.
First of all, it can be admitted that the topic chosen by the authors is welcome because many of the erythropoietin mechanisms of action have been elucidated in recent years.
My comments:
- Including figures in the manuscript would make it much easier to read and understand the fairly compiled mechanisms of action.
- The research methodology to identify and select the article should be presented in further detail reporting keywords, screened, and assessed papers.
- The manuscript would benefit from introducing an introductory statement into the review structure to guide the reader through the article.
- The unifying vision (the preliminary hypothesis of erythropoietin involvement) should be clearly defined and discussed in light of revised studies.
Author Response
GENERAL:
We thank all reviewers for detailed revision of the paper and indicated corrections, that improved our paper. All corrections suggested by reviewers including additional minor corrections by authors are marked with traced changes.
The abbreviation list of genes and protein is included in Table 2.
New references addressing tools, diseases, treatments and neocytolysis were including and reference list updated accordingly.
Reviewer 3
Comments and Suggestions for Authors
In the manuscript entitled “Molecular insights into the oxygen-sensing pathway and erythropoietin expression regulation in erythropoiesis” the authors investigate the molecular mechanisms involved in the oxygen-sensing pathway and regulation of erythropoietin production (HIF-EPO pathway) that correlate with erythropoiesis.
First of all, it can be admitted that the topic chosen by the authors is welcome because many of the erythropoietin mechanisms of action have been elucidated in recent years.
My comments:
- Including figures in the manuscript would make it much easier to read and understand the fairly compiled mechanisms of action.
As suggested by reviewer 2, we absolutely agree with this comment and we added a diagram outlining the findings of the paper at the beginning of the paper (p. 4, line 228 – 231).
- The research methodology to identify and select the article should be presented in further detail reporting keywords, screened, and assessed papers.
We agree with the comment, that the research methodology to identify and select the article should be presented in further detail, therefore describe it in the Introduction of the paper (p. 2, line 84 – 89):
“In this respect, we conducted an extensive literature search and used several in silico tools, such as Reactome [3], String [4], UniProt [5], GeneCards [6], Human Protein Atlas [7], NCBI [8], and HGNC [9] databases. The literature research was performed using keywords such as oxygen-sensing pathway, hypoxia inducible factor, HIF, HIF1A, HIF2A, EPAS1, PHD2, EGLN1, VHL, erythropoietin, EPO, EPO synthesis, hypoxia, erythropoiesis, red blood cell production, erythrocytosis. Gene names were unified according to the NCBI Gene database.”
- The manuscript would benefit from introducing an introductory statement into the review structure to guide the reader through the article.
As already suggested by reviewer 2, we explained in detail the aim of the study including the review structure at the beginning of the paper in Introduction section (p. 2, line 79 – 92):
»We reviewed factors acting on the level of (i) transcriptional regulation and (ii) protein interactions, including post-translational modifications and nuclear transport. All these factors alone or cumulatively modulate the stability or activity of the main players in the HIF-EPO pathway, and thus influence the EPO synthesis and final RBC production.«
- The unifying vision (the preliminary hypothesis of erythropoietin involvement) should be clearly defined and discussed in light of revised studies.
In aim to review all the studies, we stressed out in conclusions the role of oxygen-sensing pathway and erythropoietin in erythropoiesis regulation. The sentence is included in Conclusion section (p. 13, line 800 – 803):
“Mecanism controling HIF-EPO pathway have a prominent role in regulation of EPO synthesis, that are all together affecting the RBC production. Overall, any imbalance in HIF-EPO pathway mechanisms can result in deregulation of erythropoiesis that can lead to several diseases, including anemia, erythrocytosis and cancer.«